# Learning What To Do by Simulating the Past

**David Lindner**[*]
Department of Computer Science
ETH Zurich
`david.lindner@inf.ethz.ch`

**Rohin Shah, Pieter Abbeel & Anca Dragan**
Center for Human-Compatible AI
UC Berkeley
`{rohinmshah,pabbeel,anca}@berkeley.edu`

## Abstract

Since reward functions are hard to specify, recent work has focused on learning policies from human feedback. However, such approaches are impeded by the expense of acquiring such feedback. Recent work proposed that agents have access to a source of information that is effectively free: in any environment that humans have acted in, the state will *already* be optimized for human preferences, and thus an agent can extract information about what humans want from the state (Shah et al., 2019). Such learning is possible in principle, but requires simulating all possible past trajectories that could have led to the observed state. This is feasible in gridworlds, but how do we scale it to complex tasks? In this work, we show that by combining a learned feature encoder with learned inverse models, we can enable agents to simulate human actions backwards in time to infer what they must have done. The resulting algorithm is able to reproduce a specific skill in MuJoCo environments given a *single state* sampled from the optimal policy for that skill.

## 1 Introduction

As deep learning has become popular, many parts of AI systems that were previously designed by hand have been replaced with learned components. Neural architecture search has automated architecture design (Zoph & Le, 2017; Elsken et al., 2019), population-based training has automated hyperparameter tuning (Jaderberg et al., 2017), and self-supervised learning has led to impressive results in language modeling (Devlin et al., 2019; Radford et al., 2019; Clark et al., 2020) and reduced the need for labels in image classification (Oord et al., 2018; He et al., 2020; Chen et al., 2020). However, in reinforcement learning, one component continues to be designed by humans: the task specification. Handcoded reward functions are notoriously difficult to specify (Clark & Amodei, 2016; Krakovna, 2018), and learning from demonstrations (Ng et al., 2000; Fu et al., 2018) or preferences (Wirth et al., 2017; Christiano et al., 2017) requires a lot of human input. Is there a way that we can automate even the specification of what must be done?

It turns out that we can learn part of what the user wants *simply by looking at the state of the environment*: after all, the user will already have optimized the state towards their own preferences (Shah et al., 2019). For example, when a robot is deployed in a room containing an intact vase, it can reason that if its user wanted the vase to be broken, it would already have been broken; thus she probably wants the vase to remain intact.

However, we must ensure that the agent distinguishes between aspects of the state that the user *couldn't control* from aspects that the user *deliberately designed*. This requires us to simulate what the user *must have done* to lead to the observed state: anything that the user put effort into in the past is probably something the agent should do as well. As illustrated in Figure 1, if we observe a Cheetah balancing on its front leg, we can infer how it must have launched itself into that position. Unfortunately, it is unclear how to simulate these past trajectories that lead to the observed state. So far, this has only been done in gridworlds, where all possible trajectories can be considered using dynamic programming (Shah et al., 2019).

Our key insight is that we can sample such trajectories by starting at the observed state and simulating *backwards in time*. To enable this, we derive a gradient that is amenable to estimation through backwards simulation, and learn an inverse policy and inverse dynamics model using supervised

---

[*]Work done at the Center for Human-Compatible AI, UC Berkeley.

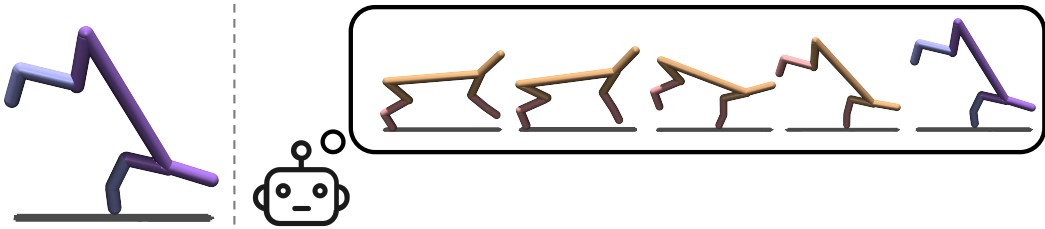

Figure 1: Suppose we observe a Cheetah balancing on its front leg (left). Given a simulator for the environment, Deep RLSP is able to infer how the cheetah must have acted to end up in this position. It can then imitate these actions in order to recreate this skill. Note that the state contains joint *velocities* in addition to positions, which makes the task more tractable than this picture might suggest.

learning to perform the backwards simulation. Then, the only remaining challenge is finding a reward representation that can be meaningfully updated from a single state observation. To that end, rather than defining the reward directly on the raw input space, we represent it as a linear combination of features learned through self-supervised representation learning. Putting these components together, we propose the *Deep Reward Learning by Simulating the Past* (Deep RLSP) algorithm.

We evaluate Deep RLSP on MuJoCo environments and show that it can recover fairly good performance on the task reward given access to a small number of states sampled from a policy optimized for that reward. We also use Deep RLSP to imitate skills generated using a skill discovery algorithm (Sharma et al., 2020), in some cases given just a *single state* sampled from the policy for that skill.

Information from the environment state cannot completely replace reward supervision. For example, it would be hard to infer how clean Bob would ideally want his room to be, if the room is currently messy because Bob is too busy to clean it. Nonetheless, we are optimistic that information from the environment state can be used to significantly reduce the burden of human supervision required to train useful, capable agents.

## 2 METHOD

In this section, we describe how Deep RLSP can learn a reward function for high dimensional environments given access only to a simulator and the observed state $s_0$.

**Notation.** A finite-horizon Markov Decision Process (MDP) $\mathcal{M} = \langle \mathcal{S}, \mathcal{A}, \mathcal{T}, r, \mathcal{P}, T \rangle$ contains a set of states $\mathcal{S}$ and a set of actions $\mathcal{A}$. The transition function $\mathcal{T} : \mathcal{S} \times \mathcal{A} \times \mathcal{S} \mapsto [0, 1]$ determines the distribution over next states given a state and an action, and $\mathcal{P}$ is a prior distribution over initial states. The reward function $r : \mathcal{S} \mapsto \mathbb{R}$ determines the agent's objective. $T \in \mathbb{Z}_+$ is a finite planning horizon. A *policy* $\pi : S \times A \mapsto [0, 1]$ specifies how to choose actions given a state. Given an initial state distribution, a policy and the transition function, we can sample a *trajectory* $\tau$ by sampling the first state from $\mathcal{P}$, every subsequent action from $\pi$, and every subsequent state from $\mathcal{T}$. We denote the probability distribution over trajectories as $\langle \mathcal{P}, \pi, \mathcal{T} \rangle$ and write $\tau \sim \langle \mathcal{P}, \pi, \mathcal{T} \rangle$ for the sampling step. We will sometimes write a single state $s$ instead of a distribution $\mathcal{P}$ if the initial state is deterministic. The goal of reinforcement learning (RL) is to find a policy $\pi^*$ that maximizes the expected cumulative reward $\mathbb{E}_{\tau \sim \langle \mathcal{P}, \pi, \mathcal{T} \rangle} \left[ \sum_{t=1}^{T} r(s_t) \right]$.

We use $\phi : \mathcal{S} \to \mathbb{R}^n$ to denote a feature function (whether handcoded or learned) that produces a feature vector of length $n$ for every state. The reward function $r$ is linear over $\phi$ if it can be expressed in the form $r(s) = \theta^T \phi(s)$ for some $\theta \in \mathbb{R}^n$.

We assume that some *past* trajectory $\tau_{-T:0} = s_{-T} a_{-T} \dots a_{-1} s_0$ produced the observed state $s_0$.

### 2.1 IDEALIZED ALGORITHM

We first explain what we would ideally do, if we had a handcoded a feature function $\phi$ and an enumerable (small) state space $\mathcal{S}$ that affords dynamic programming. This is a recap of *Reward Learning by Simulating the Past* (RLSP; Shah et al., 2019).

We assume the human follows a Boltzmann-rational policy $\pi_t(a \mid s, \theta) \propto \exp(Q_t(s, a; \theta))$, where the $Q$ values are computed using soft value iteration. Marginalizing over past trajectories, yields a distribution over the observed state $p(s_0 \mid \theta) = \sum_{s_{-T}...a_{-1}} p(\tau = s_{-T}a_{-T}...a_{-1}s_0 \mid \theta)$. We compute the maximum likelihood estimate, $\operatorname{argmax}_\theta \ln p(s_0 \mid \theta)$, via gradient ascent, by expressing the gradient of the *observed state* as a weighted combination of gradients of *consistent trajectories* (Shah et al., 2019, Appendix B):

$$\nabla_\theta \ln p(s_0 \mid \theta) = \underset{\tau_{-T:-1} \sim p(\tau_{-T:-1} \mid s_0, \theta)}{\mathbb{E}} [\nabla_\theta \ln p(\tau \mid \theta)] \tag{1}$$

$\nabla_\theta \ln p(\tau \mid \theta)$ is a gradient for inverse reinforcement learning. Since we assume a Boltzmann-rational human, this is the gradient for Maximum Causal Entropy Inverse Reinforcement Learning (MCEIRL; Ziebart et al., 2010). However, we still need to compute an expectation over all trajectories that end in $s_0$, which is in general intractable. Shah et al. (2019) use dynamic programming to compute this gradient in tabular settings.

## 2.2 Gradient as backwards-forwards consistency

**Approximating the expectation.** For higher-dimensional environments, we must approximate the expectation over past trajectories $p(\tau_{-T:-1} \mid s_0, \theta)$. We would like to sample from the distribution, but it is not clear how to sample the past conditioned on the present. Our key idea is that just as we can sample the future by rolling out forwards in time, *we should be able to sample the past by rolling out backwards in time*. Note that by the Markov property we have:

$$p(\tau_{-T:-1} \mid s_0, \theta) = \prod_{t=-T}^{-1} p(s_t \mid a_t, s_{t+1}, \ldots s_0, \theta) p(a_t \mid s_{t+1}, a_{t+1}, \ldots s_0, \theta)$$

$$= \prod_{t=-T}^{-1} p(s_t \mid a_t, s_{t+1}, \theta) p(a_t \mid s_{t+1}, \theta)$$

Thus, given the *inverse policy* $\pi_t^{-1}(a_t \mid s_{t+1}, \theta)$, the *inverse dynamics* $\mathcal{T}_t^{-1}(s_t \mid a_t, s_{t+1}, \theta)$, and the observed state $s_0$, we can sample a past trajectory $\tau_{-T:-1} \sim p(\tau_{-T:-1} \mid s_0, \theta)$ by iteratively applying $\pi^{-1}$ and $\mathcal{T}^{-1}$, starting from $s_0$. Analogous to forward trajectories, we express the sampling as $\tau_{-T:-1} \sim \langle s_0, \pi^{-1}, \mathcal{T}^{-1} \rangle$. Thus, we can write the gradient in Equation 1 as $\mathbb{E}_{\tau_{-T:-1} \sim \langle s_0, \pi^{-1}, \mathcal{T}^{-1} \rangle} [\nabla_\theta \ln p(\tau \mid \theta)]$.

**Learning $\pi$, $\pi^{-1}$ and $\mathcal{T}^{-1}$.** In order to learn $\pi^{-1}$, we must first know $\pi$. We assumed that the human was Boltzmann-rational, which corresponds to the *maximum entropy* reinforcement learning objective (Levine, 2018). We use the Soft Actor-Critic algorithm (SAC; Haarnoja et al., 2018) to estimate the policy $\pi(a \mid s, \theta)$, since it explicitly optimizes the maximum entropy RL objective.

Given the forward policy $\pi(a \mid s, \theta)$ and simulator $\mathcal{T}$, we can construct a dataset of sampled forward trajectories, and learn the inverse policy $\pi^{-1}$ and the inverse dynamics $\mathcal{T}^{-1}$ using supervised learning. Given these, we can then sample $\tau_{-T:-1}$, allowing us to approximate the expectation in the gradient. In general, both $\pi^{-1}$ and $\mathcal{T}^{-1}$ could be stochastic and time-dependent.

**Estimating the gradient for a trajectory.** We now turn to the term within the expectation, which is the inverse reinforcement learning gradient given a demonstration trajectory $\tau = s_{-T}a_{-T}\ldots s_0$. Assuming that the user is Boltzmann-rational, this is the MCEIRL gradient (Ziebart et al., 2010), which can be written as (Shah et al., 2019, Appendix A):

$$\nabla_\theta \ln p(\tau \mid \theta) = \left( \sum_{t=-T}^{0} \phi(s_t) \right) - \mathcal{F}_{-T}(s_{-T}) + \sum_{t=-T}^{-1} \left( \underset{s'_{t+1} \sim \mathcal{T}(\cdot \mid s_t, a_t)}{\mathbb{E}} \left[ \mathcal{F}_{t+1}(s'_{t+1}) \right] - \mathcal{F}_{t+1}(s_{t+1}) \right) \tag{2}$$

$\mathcal{F}$ is the expected feature count under $\pi$, that is, $\mathcal{F}_{-t}(s_{-t}) \triangleq \mathbb{E}_{\tau_{-t:0} \sim \langle s_{-t}, \pi, \mathcal{T} \rangle} \left[ \sum_{t'=-t}^{0} \phi(s_{t'}) \right]$.

The first term computes the feature counts of the demonstrated trajectory $\tau$, while the second term computes the feature counts obtained by the policy for the current reward function $\theta$ (starting from

the initial state $s_{-T}$). Since $r(s) = \theta^T \phi(s)$, these terms increase the reward of features present in the demonstration $\tau$ and decrease the reward of features under the current policy. Thus, the gradient incentivizes *consistency* between the demonstration and rollouts from the learned policy.

The last term is essentially a correction for the observed dynamics: if we see that $s_t, a_t$ led to $s_{t+1}$, it corrects for the fact that we "could have" seen some other state $s'_{t+1}$. Since this correction is zero in expectation (and expensive to compute), we drop it for our estimator.

**Gradient estimator.** After dropping the last term in Equation 2, expanding the definition of $\mathcal{F}$, and substituting in to Equation 1, our final gradient estimator is:

$$\nabla_\theta \ln p(s_0 \mid \theta) = \mathop{\mathbb{E}}_{\tau_{-T:-1} \sim \langle s_0, \pi^{-1}, \mathcal{T}^{-1} \rangle} \left[ \left( \sum_{t=-T}^{0} \phi(s_t) \right) - \mathop{\mathbb{E}}_{\tau' \sim \langle s_{-T}, \pi, \mathcal{T} \rangle} \left[ \left( \sum_{t=-T}^{0} \phi(s'_t) \right) \right] \right] \quad (3)$$

Thus, given $s_0, \theta, \pi, \mathcal{T}, \pi^{-1}$, and $\mathcal{T}^{-1}$, computing the gradient consists of three steps:

1. Simulate backwards from $s_0$, and compute the feature counts of the resulting trajectories.
2. Simulate forwards from $s_{-T}$ of these trajectories, and compute their feature counts.
3. Take the difference between these two quantities.

This again incentivizes consistency, this time between the backwards and forwards trajectories: the gradient leads to movement towards "what the human must have done" and away from "what the human would do if they had this reward". The gradient becomes zero when they are identical.

It may seem like the backwards and forwards trajectories should always be consistent with each other, since $\pi^{-1}$ and $\mathcal{T}^{-1}$ are inverses of $\pi$ and $\mathcal{T}$. The key difference is that $s_0$ imposes constraints on the backwards trajectories, but not on the forward trajectories. For example, suppose we observe $s_0$ in which a vase is unbroken, and our current hypothesis is that the user wants to break the vase. When we simulate backwards, our trajectory will contain an unbroken vase, but when we simulate forwards from $s_{-T}$, $\pi$ will break the vase. The gradient would then reduce the reward for a broken vase and increase the reward for an unbroken vase.

## 2.3 LEARNING A LATENT MDP

Our gradient still relies on a feature function $\phi$, with the reward parameterized as $r(s) = \theta^T \phi(s)$. A natural way to remove this assumption would be to instead allow $\theta$ to parameterize a neural network, which can then learn whatever features are relevant to the reward from the RLSP gradient.

However, this approach will not work because the information contained in the RLSP gradient is insufficient to identify the appropriate features to construct: after all, it is derived from a single state. If we were to learn a single unified reward using the same gradient, the resulting reward would likely be degenerate: for example, it may simply identify the observed state, that is $R(s) = \mathbb{1}[s = s_0]$.

Thus, we continue to assume that the reward is linear in features, and instead *learn the feature function* using self-supervised learning (Oord et al., 2018; He et al., 2020). In our experiments, we use a variational autoencoder (VAE; Kingma & Welling, 2014) to learn the feature function. The VAE encodes the states into a latent feature representation, which we can use to learn a reward function if the environment is fully observable, i.e., the states contain all relevant information.

For partially observable environments recurrent state space models (RSSMs; Karl et al., 2017; Doerr et al., 2018; Buesing et al., 2018; Kurutach et al., 2018; Hafner et al., 2019; 2020) could be used instead. These methods aim to learn a *latent MDP*, by computing the states using a recurrent model over the observations, thus allowing the states to encode the history. For such a model, we can imagine that the underlying POMDP has been converted into a latent MDP whose feature function $\phi$ is the identity. We can then compute gradients directly in this latent MDP.

## 2.4 DEEP RLSP

Putting these components together gives us the Deep RLSP algorithm (Algorithm 1). We first learn a feature function $\phi$ using self-supervised learning, and then train an inverse dynamics model $\mathcal{T}^{-1}$, all using a dataset of environment interactions (such as random rollouts). Then, we update $\theta$ using

---

**Algorithm 1** The DEEP RLSP algorithm. The initial dataset of environment interactions $D$ can be constructed in many different ways: random rollouts, human play data, curiosity-driven exploration, etc. The specific method will determine the quality of the learned features.

---

**procedure** DEEP RLSP($\{s_0\}, \mathcal{T}$)
 $D \leftarrow$ dataset of environment interactions
 Initialize $\phi_e, \phi_d, \pi, \pi^{-1}, \mathcal{T}^{-1}, \theta$ randomly.
 $\phi_e, \phi_d \leftarrow$ SelfSupervisedLearning($D$)       ▷ Train encoder and decoder for latent MDP
 Initialize experience replay $E$ with data in $D$.
 $\mathcal{T}^{-1} \leftarrow$ SupervisedLearning($D$)           ▷ Train inverse dynamics
 $T \leftarrow 1$                  ▷ Start horizon at 1
 **for** $i$ in [1..num_epochs] **do**
  $\pi \leftarrow$ SAC($\theta$)               ▷ Train policy
  $\pi^{-1} \leftarrow$ SupervisedLearning($\phi_e, E$)      ▷ Train inverse policy
  $\theta \leftarrow \theta + \alpha \times$ COMPUTEGRAD($\{s_0\}, \pi, \mathcal{T}, \pi^{-1}, \mathcal{T}^{-1}, T, \phi_e$)    ▷ Update $\theta$
  **if** gradient magnitudes are sufficiently low **then**
   $T \leftarrow T + 1$             ▷ Advance horizon
 **return** $\theta, \phi_e$
**procedure** COMPUTEGRAD($\{s_0\}, \pi, \mathcal{T}, \pi^{-1}, \mathcal{T}^{-1}, T, \phi_e$)
 $\{\tau_{\text{backward}}\} \leftarrow$ Rollout($\{s_0\}, \pi^{-1}, \mathcal{T}^{-1}, T$)      ▷ Simulate backwards from $s_0$
 $\phi_{\text{backward}} \leftarrow$ AverageFeatureCounts($\phi_e, \{\tau_{\text{backward}}\}$)    ▷ Compute backward feature counts
 $\{s_{-T}\} \leftarrow$ FinalStates($\{\tau_{\text{backward}}\}$)
 $\{\tau_{\text{forward}}\} \leftarrow$ Rollout($\{s_{-T}\}, \pi, \mathcal{T}, T$)       ▷ Simulate forwards from $s_{-T}$
 $\phi_{\text{forward}} \leftarrow$ AverageFeatureCounts($\phi_e, \{\tau_{\text{forward}}\}$)    ▷ Compute forward feature counts
 Relabel $\{\tau_{\text{backward}}\}, \{\tau_{\text{forward}}\}$ and add them to $E$.
 **return** $\phi_{\text{backward}} - \phi_{\text{forward}}$

---

Equation 3, and continually train $\pi$, and $\pi^{-1}$ alongside $\theta$ to keep them up to date. The full algorithm also adds a few bells and whistles that we describe next.

**Initial state distribution $\mathcal{P}$.** The attentive reader may wonder why our gradient appears to be independent of $\mathcal{P}$. This is actually not the case: while $\pi$ and $\mathcal{T}$ are independent of $\mathcal{P}$, $\pi^{-1}$ and $\mathcal{T}^{-1}$ *do* depend on it. For example, if we observe Alice exiting the San Francisco airport, the corresponding $\pi^{-1}$ should hypothesize different flights if she started from New York than if she started from Tokyo.

However, in order to actually produce such explanations, we must train $\pi^{-1}$ and $\mathcal{T}^{-1}$ solely on trajectories of length $T$ starting from $s_{-T} \sim \mathcal{P}$. We instead train $\pi^{-1}$ and $\mathcal{T}^{-1}$ on a variety of trajectory data, which loses the useful information in $\mathcal{P}$, but leads to several benefits. First, we can train the models on exactly the distributions that they will be used on, allowing us to avoid failures due to distribution shift. Second, the horizon $T$ is no longer critical: previously, $T$ encoded the separation in time between $s_{-T}$ and $s_0$, and as a result misspecification of $T$ could cause bad results. Since we now only have information about $s_0$, it doesn't matter much what we set $T$ to, and as a result we can use it to set a curriculum (discussed next). Finally, this allows Deep RLSP to be used in domains where an initial state distribution is not available.

Note that we are no longer able to use information about $\mathcal{P}$ through $\pi^{-1}$ and $\mathcal{T}^{-1}$. However, having information about $\mathcal{P}$ might be crucial in some applications to prevent Deep RLSP from converging to a degenerate solution with $s_{-T} = s_0$ and a policy $\pi$ that does nothing. While we did not find this to be a problem in our experiments, we discuss a heuristic to incorporate information about $s_{-T}$ into Deep RLSP in Appendix C.

**Curriculum.** Since the horizon $T$ is no longer crucial, we can use it to provide a curriculum. We initially calculate gradients with low values of $T$, to prevent compounding errors in our learned models, and making it easier to enforce backwards-forwards consistency, and then slowly grow $T$, making the problem harder. In practice, we found this crucial for performance: intuitively, it is much easier to make short backwards and forwards trajectories consistent than with longer trajectories; the latter would likely have much higher variance.

**Multiple input states.** If we get multiple independent $s_0$ as input, we average their gradients.

**Experience replay.** We maintain an experience replay buffer $E$ that persists across policy training steps. We initialize $E$ with the same set of environment interactions that the feature function and inverse dynamics model are trained on. When computing the gradient, we collect all backward and forward trajectories and add them to $E$. To avoid compounding errors from the inverse dynamics model, we relabel all transitions using a simulator of the environment. Whenever we'd add a transition $(s, a, s')$ to $E$, we initialize the simulator at $s$ and execute $a$ to obtain $\tilde{s}$ and add transition $(s_1, a, \tilde{s})$ to $E$ instead.

# 3 EXPERIMENTS

## 3.1 SETUP

To demonstrate that Deep RLSP can be scaled to complex, continuous, high-dimensional environments, we use the MuJoCo physics simulator (Todorov et al., 2012). We consider the *Inverted Pendulum*, *Half-Cheetah* and *Hopper* environments implemented in Open AI Gym (Brockman et al., 2016). The hyperparameters of our experiments are described in detail in Appendix B. We provide code to replicate our experiments at `https://github.com/HumanCompatibleAI/deep-rlsp`.

**Baselines.** To our knowledge, this is the first work to train policies using a single state as input. Due to lack of alternatives, we compare against GAIL (Ho & Ermon, 2016) using the implementation from the `imitation` library (Wang et al., 2020). For each state we provide to Deep RLSP, we provide a transition $(s, a, s')$ to GAIL.

**Ablations.** In Section 2.2, we derived a gradient for Deep RLSP that enforces consistency between the backwards and forwards trajectories. However, we could also ignore the temporal information altogether. If an optimal policy led to the observed state $s_0$, then it is probably a good bet that $s_0$ is high reward, and that the agent should try to keep the state similar to $s_0$. Thus, we can simply set $\theta = \frac{\phi(s_0)}{||\phi(s_0)||}$, and not deal with $\pi^{-1}$ and $\mathcal{T}^{-1}$ at all.

How should we handle multiple states $s_0^1, \ldots, s_0^N$? Given that these are all sampled i.i.d. from rollouts of an optimal policy, a natural choice is to simply average the feature vectors of all of the states, which we call *AverageFeatures*. Alternatively, we could view each of the observed states as a potential *waypoint* of the optimal policy, and reward an agent for being near any one of them. We implement this *Waypoints* method as $R(s) = \max_i \frac{\phi(s_0^i)}{||\phi(s_0^i)||} \cdot \phi(s)$. Note that both of these ablations still require us to learn the feature function $\phi$.

**Feature learning dataset.** By default, we use random rollouts to generate the initial dataset that is used to train the features $\phi$ and the inverse model $\mathcal{T}^{-1}$. (This is $D$ in Algorithm 1.) However, in the inverted pendulum environment, the pendulum falls very quickly in random rollouts, and $\mathcal{T}^{-1}$ never learns what a balanced pendulum looks like. So, for this environment only, we combine random rollouts with rollouts from an expert policy that balances the pendulum.

## 3.2 GRIDWORLD ENVIRONMENTS

As a first check, we consider the gridworld environments in Shah et al. (2019). In these stylized gridworlds, self-supervised learning should not be expected to learn the necessary features. For example, in the room with vase environment, the two door features are just particular locations, with no distinguishing features that would allow self-supervised learning to identify these locations as important. So, we run Algorithm 1 without the feature learning and instead use the pre-defined feature function of the environments. With this setup we are able to use Deep RLSP to recover the desired behavior from a single state in all environments in which the exact RLSP algorithm is able to recover it. However, AverageFeatures fails on several of the environments. Since only one state is provided, Waypoints is equivalent to AverageFeatures. It is not clear how to apply GAIL to these environments, and so we do not compare to it. Further details on all of the environments and results can be found in Appendix A.

| Environment | SAC | # states | Deep RLSP | AverageFeatures | Waypoints | GAIL |
|---|---|---|---|---|---|---|
| Inverted Pendulum | 1000 | 1 | 303 (299) | 6 (2) | N/A | **1000 (0)** |
| | | 10 | 335 (333) | 3 (1) | 4 (1) | **1000 (0)** |
| | | 50 | 339 (331) | 6 (4) | 3.7 (0.3) | **1000 (0)** |
| Cheetah (forward) | 13236 | 1 | **4591 (2073)** | **6466 (3343)** | N/A | -288 (55) |
| | | 10 | **6917 (421)** | **6245 (2352)** | -10 (23) | -296 (172) |
| | | 50 | **6078 (589))** | 4504 (2970) | -126 (38) | -54 (295) |
| Cheetah (backward) | 13361 | 1 | 5730 (2733) | **12443 (645)** | N/A | -335 (46) |
| | | 10 | 7917 (249) | **12829 (651)** | -80 (388) | -283 (45) |
| | | 50 | 7588 (171) | **11616 (178)** | -509 (87) | 2113 (1015) |
| Hopper (terminate) | 3274 | 1 | 68 (8) | 99 (45) | N/A | **991 (9)** |
| | | 10 | 47 (21) | 159 (126) | 58 (7) | **813 (200)** |
| | | 50 | 72 (1) | 65 (36) | 14 (4) | **501 (227)** |
| Hopper (penalty) | 3363 | 1 | **1850 (634)** | **2537 (363)** | N/A | 990 (9) |
| | | 10 | **2998 (62)** | **3103 (64)** | 709 (133) | 784 (229) |
| | | 50 | **1667 (737)** | **2078 (581)** | **1612 (785)** | 508 (259) |

Table 1: Average returns achieved by the policies learned through various methods, for different numbers of input states. The states are sampled from a policy trained using SAC on the true reward function; the return of that policy is given as a comparison. Besides the SAC policy return, all values are averaged over 3 seeds and the standard error is given in parentheses. We don't report Waypoints on 1 state as it is identical to AverageFeatures on 1 state.

## 3.3 SOLVING THE ENVIRONMENTS WITHOUT ACCESS TO THE REWARD FUNCTION

First we look at the typical target behavior in each environment: balancing the inverted pendulum, and making the half-cheetah and the hopper move forwards. Additionally we consider the goal of making the cheetah run backwards (that is, the negative of its usual reward function). We aim to use Deep RLSP to learn these behaviors *without having access to the reward function*.

We train a policy using soft actor critic (SAC; Haarnoja et al., 2018) to optimize for the true reward function, and sample either 1, 10 or 50 states from rollouts of this policy to use as input. We then use Deep RLSP to infer a reward and policy. Ideally we would evaluate this learned policy rather than reoptimizing the learned reward, since learned reward models can often be gamed (Stiennon et al., 2020), but it would be too computationally expensive to run the required number of SAC steps during each policy learning step. As a result, we run SAC for many more iterations on the inferred reward function, and evaluate the resulting policy on the true reward function (which Deep RLSP does not have access to).

Results are shown in Table 1. In Hopper, we noticed that videos of the policies learned by Deep RLSP looked okay, but the quantitative evaluation said otherwise. It turns out that the policies learned by Deep RLSP do jump, as we might want, but they often fall down, terminating the episode; in contrast GAIL policies stand still or fall over slowly, leading to later termination and explaining their better quantitative performance. We wanted to also evaluate the policies without this termination bias, and so we evaluate the same policies in an environment that does not terminate the episode, but provides a negative reward instead; in this evaluation both Deep RLSP and AverageFeatures perform much better. We also provide videos of the learned policies at `https://sites.google.com/view/deep-rlsp`, which show that the policies learned by Deep RLSP do exhibit hopping behavior (though with a strong tendency to fall forward).

GAIL is only able to learn a truly good policy for the (very simple) inverted pendulum, even though it gets states and actions as input. Deep RLSP on the other hand achieves reasonable behavior (though clearly not expert behavior) in all of the environments, using only states as input. Surprisingly, the AverageFeatures method also performs quite well, even beating the full algorithm on some tasks, though failing quite badly on Pendulum. It seems that the task of running forward or backward is very well specified by a single state, since it can be inferred even without any information about the dynamics (except that which is encoded in the features learned from the initial dataset).

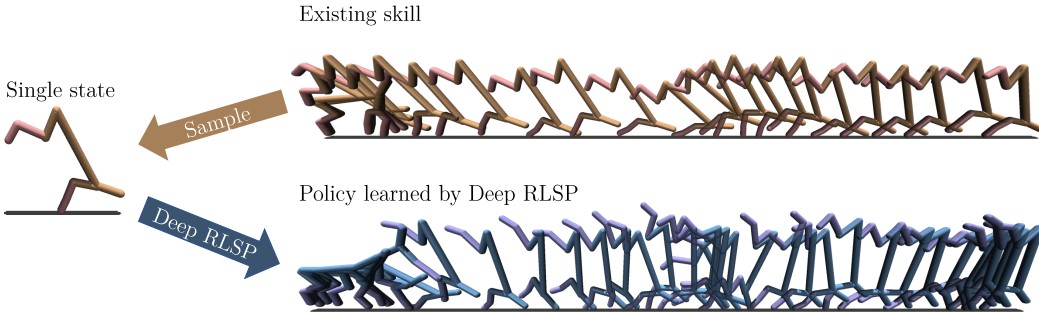

Figure 2: We sample a few states from a policy performing a specific skill to provide as input. Here, Deep RLSP learns to balance the cheetah on the front leg from a *single state*. We provide videos of the original skills and learned policies at: `https://sites.google.com/view/deep-rlsp`.

## 3.4    LEARNING SKILLS FROM A SINGLE STATE

We investigate to what extent Deep RLSP can learn other skills where the reward is not clear. Evaluation on these tasks is much harder, because there is no ground truth reward. Therefore we evaluate qualitatively how similar the policies learned by Deep RLSP are to the original skill. We also attempted to quantify similarity by checking how quickly a discriminator could learn to distinguish between the learned policy and the original skill, but unfortunately this metric was not conclusive (results are reported in Appendix D.1). Unlike the previous case, we do not reoptimize the learned reward and only look at the policies learned by Deep RLSP.

We consider skills learned by running Dynamics-Aware Unsupervised Discovery of Skills (DADS; Sharma et al., 2020). Since we are not interested in navigation, we remove the "x-y prior" used to get directional skills in DADS. We run DADS on the half-cheetah environment and select all skills that are not some form of running. This resulted in two skills: one in which the cheetah is moving forward making big leaps (*"jumping"*) and one in which it is slowly moving forward on one leg (*"balancing"*). As before we roll out these policies and sample individual states from the trajectories to provide as an input for Deep RLSP. We then evaluate the policy learned by Deep RLSP. Since the best evaluation here is to simply watch what the learned policy does, we provide videos of the learned policies at `https://sites.google.com/view/deep-rlsp`. We also provide visualizations in Appendix D.2.

The first thing to notice is that relative to the ablations, only Deep RLSP is close to imitating the skill. None of the other policies resemble the original skills at all. While AverageFeatures could perform well on simple tasks such as running, the full algorithm is crucial to imitate more complex behavior.

Between Deep RLSP and GAIL the comparison is less clear. Deep RLSP can learn the balancing skill fairly well from a single state, which we visualize in Figure 2 (though we emphasize that the videos are much clearer). Like the original skill, the learned policy balances on one leg and slowly moves forward by jumping, though with slightly more erratic behavior. However, the learned policy sometimes drops back to its feet or falls over on its back. We suspect this is an artifact of the short horizon ($T \leq 10$) used for simulating the past in our algorithm. A small horizon is necessary to avoid compounding errors in the learned inverse dynamics model, but can cause the resulting behavior to be more unstable on timescales greater than $T$. We see similar behavior when given 10 or 50 states. GAIL leads to a good policy given a single transition, where the cheetah balances on its front leg and head (rather than just the front leg), but does not move forward very much. However, with 10 or 50 transition, the policies learned by GAIL do not look at all like balancing.

However, the jumping behavior is harder to learn, especially from a single state. We speculate that here a single state is less informative than the balancing state. In the balancing state, the low joint velocities tell us that the cheetah is not performing a flip, suggesting that we had optimized for this specific balancing state. On the other hand, with the jumping behavior, we only get a single state of the cheetah in the air with high velocity, which is likely not sufficient to determine what the jump looked like exactly. In line with this hypothesis, at 1 state Deep RLSP learns to erratically hop, at 10 states it executes slightly bigger jumps, and at 50 states it matches the original skill relatively closely.

The GAIL policies for jumping are also reasonable, though in a different way that makes it hard to compare. Using 1 or 10 transitions, the policy doesn't move very much, staying in contact with the ground most of the time. However, at 50 transitions, it performs noticeably forward hops slightly smoother than the policy learned by Deep RLSP.

## 4 RELATED WORK

**Learning from human feedback.** Many algorithms aim to learn good policies from human demonstrations, including ones in imitation learning (Ho & Ermon, 2016) and inverse reinforcement learning (IRL; Ng et al., 2000; Abbeel & Ng, 2004; Fu et al., 2018). Useful policies can also be learned from other types of feedback, such as preferences (Christiano et al., 2017), corrections (Bajcsy et al., 2017), instructions (Bahdanau et al., 2019), or combinations of feedback modalities (Ibarz et al., 2018).

While these methods require expensive human feedback, Deep RLSP instead *simulates* the trajectories that must have happened. This is reflected in the algorithm: in Equation 1, the inner gradient corresponds to an inverse reinforcement learning problem. While we used the MCEIRL formulation (Ziebart et al., 2010), other IRL algorithms could be used instead (Fu et al., 2018).

**Learning from observations.** For many tasks, we have demonstrations *without action labels*, e.g., YouTube videos. Learning from Observations (LfO; Torabi et al., 2019; Gandhi et al., 2019) aims to recover a policy from such demonstrations. Similarly to LfO, we do not have access to action labels, but our setting is further restricted to observing only a small number of states.

## 5 LIMITATIONS AND FUTURE WORK

**Summary.** Learning useful policies with neural networks requires significant human effort, whether it is done by writing down a reward function by hand, or by learning from explicit human feedback such as preferences or demonstrations. We showed that it is possible to reduce this burden by extracting "free" information present in the current state of the environment. This enables us to imitate policies in MuJoCo environments with access to just a few states sampled from those policies. We hope that Deep RLSP will help us train agents that are better aligned with human preferences.

**Learned models.** The Deep RLSP gradient depends on having access to a good model of $\pi$, $\mathcal{T}$, $\pi^{-1}$, and $\mathcal{T}^{-1}$. In practice, it was quite hard to train sufficiently good versions of the inverse models. This could be a significant barrier to practical implementations of Deep RLSP. It can also be taken as a sign for optimism: self-supervised representation learning through deep learning is fairly recent and is advancing rapidly; such advances will likely translate directly into improvements in Deep RLSP.

**Computational cost.** Imitation learning with full demonstrations can already be quite computationally expensive. Deep RLSP learns several distinct neural network models, and then *simulates* potential demonstrations, and finally imitates them. Unsurprisingly, this leads to increased computational cost.

**Safe RL.** Shah et al. (2019) discuss how the exact RLSP algorithm can be used to avoid negative side-effects in RL by combining preferences learned from the initial state with a reward function. While we focused on learning hard to specify behavior, Deep RLSP can also be used to learn to avoid negative side-effects, which is crucial for safely deploying RL systems in the real world (Amodei et al., 2016).

**Multiagent settings.** In any realistic environment, there is not just a single "user" who is influencing the environment: many people act simultaneously, and the state is a result of joint optimization by all of them. However, our model assumes that the environment state resulted from optimization by a single agent, which will not take into account the fact that each agent will have constraints imposed upon them by other agents. We will likely require new algorithms for such a setting.

### ACKNOWLEDGMENTS

This work was partially supported by Open Philanthropy, AFOSR, ONR YIP, NSF CAREER, NSF NRI, and Microsoft Swiss JRC. We thank researchers at the Center for Human-Compatible AI and the InterACT lab for helpful discussion and feedback.

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

## A  GRIDWORLD ENVIRONMENTS

Here we go into more detail on the experiments in Section 3.2, in which we ran Deep RLSP on the environment suite constructed in Shah et al. (2019).

In this test suite, each environment comes equipped with an *observed state* $s_0$, an *initial state* $s_{-T}$, a specified reward $R_{\text{spec}}$, and a true reward $R_{\text{true}}$. A given algorithm should be run on $s_0$ and optionally also $s_{-T}$ and produce an inferred reward $R_{\text{inferred}}$. This is then added to the specified reward to produce $R_{\text{final}} = R_{\text{spec}} + \lambda R_{\text{inferred}}$, where $\lambda$ is a hyperparameter that determines the weighting between the two. An optimal policy for $R_{\text{final}}$ is then found using value iteration, and the resulting policy is evaluated according to $R_{\text{true}}$.

There is no clear way to set $\lambda$: it depends on the scales of the rewards. We leverage the fact that $R_{\text{spec}}$ is deliberately chosen to incentivize bad behavior, such that we know $\lambda = 0$ will always give incorrect behavior. So, we normalize $R_{\text{inferred}}$, and then increase $\lambda$ from 0 until the behavior displayed by the final policy changes.

Since GAIL does not produce a reward function as output, we do not run it here. We do however report results with AverageFeatures (which is equivalent to Waypoints here, because there is only a single observed state).

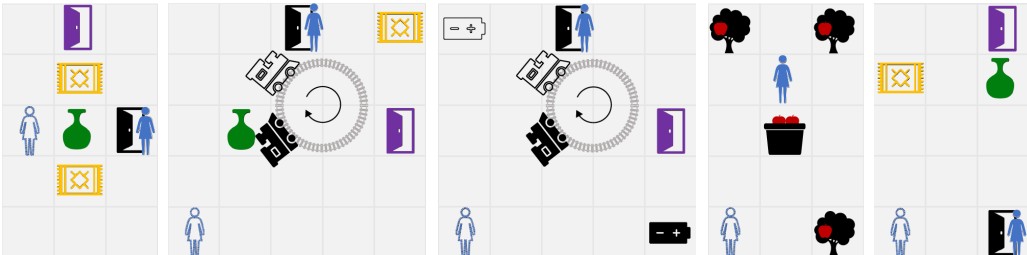

Figure 3: Reproduction of part of Figure 2 in Shah et al. (2019) illustrating the gridworld environments that we test on.

From left to right, the environments are:

1. Room with vase: $R_{\text{spec}}$ has weight 1 for the purple door feature, and 0 for all other weights. $R_{\text{true}}$ additionally has weight -1 for the broken vases feature. Since we observe a state in which the vase is unbroken, we can infer that the human avoided breaking the vase, and so that there should be a negative weight on broken vases. Deep RLSP indeed does this and so avoids breaking the vase. AverageFeatures fails to do so, though this is due to a quirk in the feature encoding. In particular, the feature counts the number of broken vases, and so the inferred $\theta$ has a value of zero for this feature, effectively ignoring it. If we change the featurization to instead count the number of *unbroken* vases, then AverageFeatures would likely get the right behavior.

2. Toy train: In this environment, we observe a state in which an operational train is moving around a track. Once again, $R_{\text{spec}}$ just has weight 1 on the purple door feature. $R_{\text{true}}$ additionally has weight -1 on broken vases and trains. Deep RLSP appropriately avoids breaking objects, but AverageFeatures does not.

3. Batteries: We observe a state in which the human has put a battery in the train to keep it operational ($s_{-T}$ has two batteries while $s_0$ only has one). $R_{\text{spec}}$ still has weight 1 on the purple door feature. $R_{\text{true}}$ additionally has weight -1 on allowing the train to run out of power. Algorithms should infer that it is good to put batteries in the train to keep it operational, even though this irreversibly uses up the battery. Deep RLSP correctly does this, while AverageFeatures does not. In fact, AverageFeatures incorrectly infers that batteries should *not* be used up.

4. Apples: We observe a state in which the human has collected some apples and placed them in a basket. $R_{\text{spec}}$ is always zero, while $R_{\text{true}}$ has weight 1 on the number of apples in the basket. The environment tests whether algorithms can infer that it is good for there to be apples in the basket. Deep RLSP does this, learning a policy that continues to collect apples

and place them in the basket. AverageFeatures also learns to place apples in the basket, but does not do so as effectively as Deep RLSP, because AverageFeatures also rewards the agent for staying in the original location, leading it to avoid picking apples from the tree that is furthest away.

5. Room with far away vase: This is an environment that aims to show what *can't* be learned: in this case, the breakable vase is so far away, that it is not much evidence that the human has not broken it so far. As a result, algorithms should *not* learn anything significant about whether or not to break vases. This is indeed the case for Deep RLSP, as well as AverageFeatures (though once again, in the latter case, this is dependent on the specific form of the feature).

Overall, Deep RLSP has the same behavior on these environments as RLSP, while AverageFeatures does not.

## B Architecture and hyperparameter choices

In this section we describe the architecture choices for the models used in our algorithm and the hyperparameter choices in our experiments. All models are implemented using the TensorFlow framework.

### B.1 Feature function

We use a variational autoencoder (VAE; Kingma & Welling, 2014) to learn the feature function. The encoder and decoder consist of 3 feed-forward layers of size 512. The latent space has dimension 30. The model is trained for 100 epochs on 100 rollouts of a random policy in the environment. During training we use a batch size of 500 and a learning rate of $10^{-5}$. We use the standard VAE loss function, but weight the KL-divergence term with a factor $c = 0.001$, which reduces the regularization and empirically improved the reconstruction of the model significantly. We hypothesize that the standard VAE regularizes too much in our setting, because the latent space has a higher dimension than the input space, which is not the case in typical dimensionality reduction settings.

### B.2 Inverse dynamics model

Our inverse dynamics model is a feed-forward neural network with 5 layers of size 1024 with ReLU activations. We train it on 1000 rollouts of a random policy in the environment for 100 epochs, with a batch size of 500 and a learning rate of $10^{-5}$.

Note that the model predicts the previous observation given the current observation and action; it does not use the feature representation. We found the model to perform better if it predicts the residual $o_{t-1} - o_t$ given $o_t$ and $a_t$ instead of directly predicting $o_{t-1}$.

We normalize all inputs to the model to have zero mean and unit variance. To increase robustness, we also add zero-mean Gaussian noise with standard deviation 0.001 to the inputs and labels during training and clip the outputs of the model to the range of values observed during training.

### B.3 Policy

For learning the policy we use the *stable-baselines* implementation of Soft Actor-Critic (SAC) with its default parameters for the MuJoCo environments (Haarnoja et al., 2018; Hill et al., 2018). Each policy update during Deep RLSP uses $10^4$ total timesteps for the cheetah, $2 \times 10^4$ for the hopper. We perform the policy updates usually starting from the last iteration's policy, except in the pendulum environment, where we randomly initialize the policy in each iteration and train it using $5 \times 10^4$ iterations of SAC. We evaluate the final reward function generally using $2 \times 10^6$ timesteps, except for the pendulum, where we use $6 \times 10^4$.

### B.4 Inverse policy

Because the inverse policy is not deterministic, we represent it with a *mixture density network*, a feed-forward neural network that outputs a mixture of Gaussian distributions (Bishop, 1994).

| Environment | SAC | # states | Deep RLSP (no gradient weights) | Deep RLSP (with gradient weights) |
|---|---|---|---|---|
| Inverted Pendulum | 1000 | 1 | **303 (299)** | 6 (3) |
| | | 10 | **335 (333)** | **667 (333)** |
| | | 50 | **339 (331)** | 5 (3) |
| Cheetah (forward) | 13236 | 1 | **4591 (2073)** | **4833 (2975)** |
| | | 10 | **6917 (421)** | **6299 (559)** |
| | | 50 | 6078 (589) | **7657 (177)** |
| Cheetah (backward) | 13361 | 1 | **5730 (2733)** | **5694 (2513)** |
| | | 10 | **7917 (249)** | **8102 (624)** |
| | | 50 | **7588 (171)** | **7795 (551)** |
| Hopper (terminate) | 3274 | 1 | 68 (8) | 70 (33) |
| | | 10 | 47 (21) | 81 (9) |
| | | 50 | 72 (1) | 81 (15) |
| Hopper (penalty) | 3363 | 1 | **1850 (634)** | 1152 (583) |
| | | 10 | **2998 (62)** | 1544 (608) |
| | | 50 | **1667 (737)** | **2020 (571)** |

Table 2: Ablation of the gradient weighting heuristic described in Section 2.4. We report average returns (over 3 random seeds) achieved by the policies learned with and without the heuristic, for different numbers of input states. Experiment setup is the same as in Table 1.

The network has 3 layers of size 512 with ReLU activations and outputs a mixture of 5 Gaussians with a fixed variance of 0.05.

To update the inverse policy we sample batches with batch size 500 from the experience replay, apply the forward policy and the forward transition model on the states to label the data. We then train the model with a learning rate of $10^{-4}$.

### B.5 DEEP RLSP HYPERPARAMETERS

We run Deep RLSP with a learning rate of 0.01, and use 200 forward and backward trajectories to estimate the gradients. Starting with $T = 1$ we increment the horizon when the gradient norm drops below 2.0 or after 10 steps, whichever comes first. We run the algorithm until $T = 10$.

## C    HEURISTIC FOR INCORPORATING INFORMATION ABOUT THE INITIAL STATE

In Section 2.4 we discussed that it might be necessary for Deep RLSP to have information about the distribution $\mathcal{P}$ of the initial state $s_{-T}$. Since in our setup Deep RLSP can not obtain any information about $\mathcal{P}$ through $\pi^{-1}$ and $\mathcal{T}^{-1}$, here we present a heuristic to incorporate the information elsewhere.

Specifically, we weight every backwards trajectory by the cosine similarity between the final state $s_{-T}$, and a sample $\hat{s}_{-T} \sim \mathcal{P}$. This weights gradient terms higher that correspond to trajectories that are more likely given our knowledge about $\mathcal{P}$ and weights trajectories lower that end in a state $s_{-T}$ that has low probability under $\mathcal{P}$.

To test whether this modification improves the performance of Deep RLSP, we compared Deep RLSP with this gradient weighting heuristic to Deep RLSP without it as it was presented in the main paper.

First, we ran Deep RLSP with the gradient weighting on the gridworld environments from Shah et al. (2019), described in Section 3.2 and Appendix A. The results are identical to the case when using the heuristics.

Next, we tested on the tasks in the MuJoCo environments described in Section 3.3. We report the results in Table 2, alongside the previously reported results without the gradient weighting. The results are quite similar, suggesting that the gradient weighting does not make much of a difference in these environments.

# D  ANALYSIS OF THE LEARNED SKILLS

## D.1  TRAINING A DISCRIMINATOR

In the main text, we focused on visual evaluation of the learned skills, because it is difficult to define a metric that properly measures the similarity between an original skill and one learned by Deep RLSP. In this section, we attempt to quantify the similarity between policies by training a discriminator to distinguish trajectories from the policies. Conceptually, the easier it is to train this discriminator, the more different the two policies are. We could thus use this to check how similar our learned policies are to the original skills.

We train a neural network with a single hidden layer of size 10 with ReLU activation functions. We sample trajectories from both policies and randomly sample trajectory pieces consisting of 5 observations to train the model on. We label the trajectory pieces with a binary label depending on which policy they come from, and then use a cross-entropy loss to train the model. To ensure comparable results, we keep this setup the same for all policies and average the resulting learning curves over 10 different random seeds.

The resulting learning curves are shown in fig. 4. The differences between the learning curves are relatively small overall, suggesting that we cannot draw strong conclusions from this experiment. In addition, while the AverageFeatures and Waypoints ablations can be seen to be extremely bad visually relative to GAIL and Deep RLSP, this is not apparent from the learning curves. As a result, we conclude that this is not actually a good metric to judge performance. (Note that if we were to use the metric, it would suggest that Deep RLSP is best for the balancing learning skill, while for the jumping skill GAIL is better for 1 and 50 states and Deep RLSP is better for 10 states.)

## D.2  VISUALIZATION OF LEARNED SKILLS

Here we provide larger visualizations of the skills learned in the experiments discussed in Section 3.4 of the main paper. For each experiment we show the original policy, the states sampled from this policy and given as an input to Deep RLSP, the policy learned by the AverageFeatures ablation, and the policy learned by Deep RLSP in  figs. 5 to 10 (on future pages). Again, we emphasize that the visual comparison is easier with videos of the policies which we provide at `https://sites.google.com/view/deep-rlsp` (including Waypoints and AverageFeatures ablations).

# E  THINGS WE TRIED THAT DID NOT WORK

Here we list a few variations of the Deep RLSP algorithm that we tested on the MuJoCo environments that failed to provide good results.

- We tried to learn a latent state-space jointly with a latent dynamics model using a recurrent state-space model (RSSM). However, we found existing models too brittle to reliably learn a good dynamics model. The reward function and policy learned by Deep RLSP worked in the RSSM but did not generalize to the actual environment.

- We also tried learning a forward dynamics model from the initial set of rollouts, similarly to how we learn an inverse dynamics model, rather than relying on the simulator $\mathcal{T}$. However, we found this to cause a similar issue as the RSSM: the reward function and policy learned by Deep RLSP did not generalize to the actual environment. However, we hope that progress in model-based RL will allow us to implement Deep RLSP using only learned dynamics models in the future.

- Using an mixture density network instead of an MLP to model the inverse dynamics did not improve the performance of the algorithm. We suspect this to be because in the MuJoCo simulator the dynamics and the inverse dynamics are "almost deterministic".

- Updating the inverse dynamics model and the feature function during Deep RLSP by training it on data from the experience replay did not improve performance and in some cases significantly decreased performance. The decrease in performance seems to have been caused by the feature function changing too much and the training of the other models suffering from catastrophic forgetting as a result.

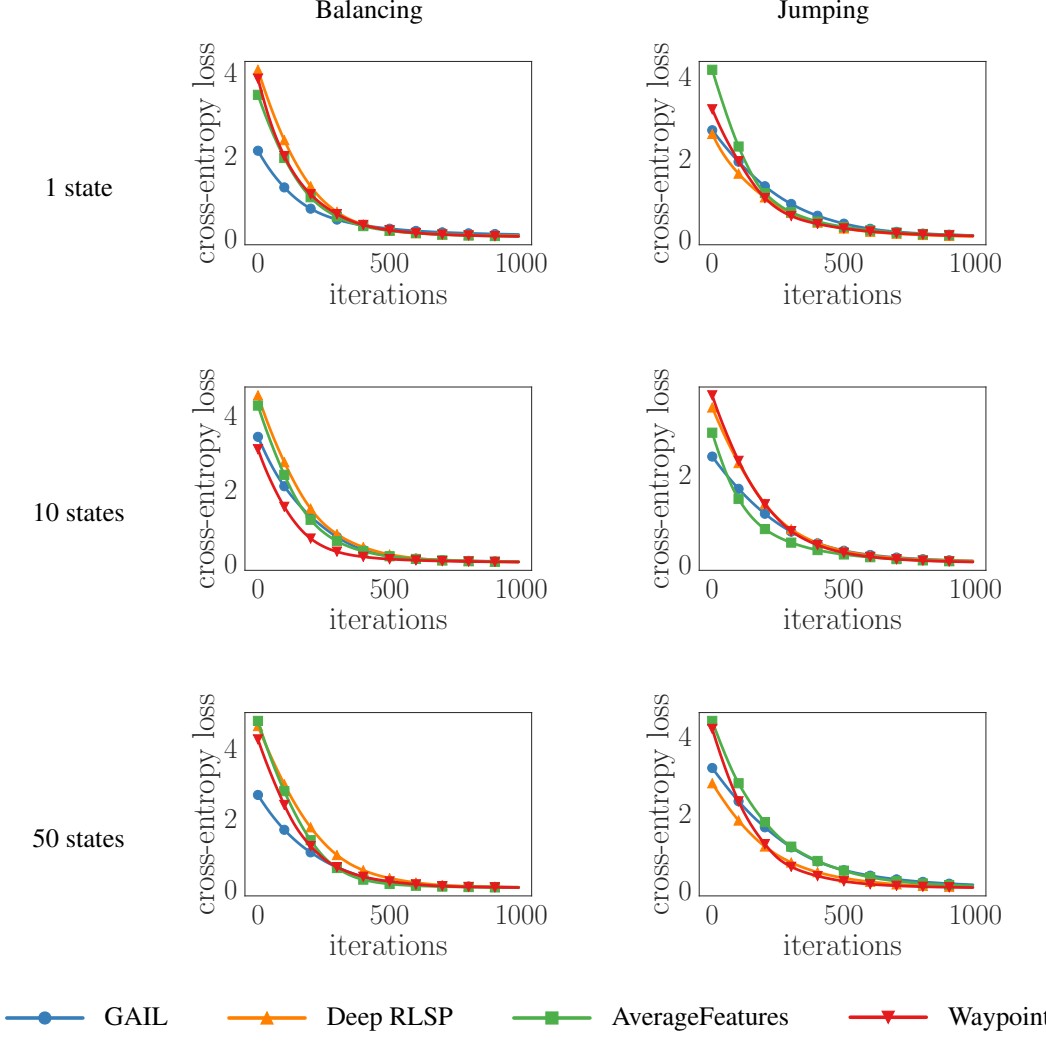

Figure 4: Learning curves for training a discriminator to distinguish the learned skill from the original skill averaged over 10 random seeds. A slower learning curve indicates that the learned skill is more similar to the original skill, that is, higher is better.

- In the main paper we evaluated the policies learned by Deep RLSP from jumping and balancing skills. However, we also looked at policies obtained by optimizing for the learned reward. These also showed similarities to the original skills but they were significantly worse then the policies directly learned by Deep RLSP. For the jumping skill the optimized policies jump very erratically, and for the balancing skill they tend to fall over or perform forward flips. This discrepency is a result of the policy updates during Deep RLSP only using a limited number of iterations. It seems like in these experiments the learned reward functions lead to good policies when optimized for weakly but do not produce good policies when optimized for strongly. We saw in preliminary experiments that increasing the number of iterations for updating the policies during Deep RLSP reduces this discrepency. However, the resulting algorithm was computationally too expensive to evaluate with our resources.

- We tried running Deep RLSP for longer horizons up to $T = 30$, but found the results to be worse than for $T = 10$ which we reported in the main paper. We hypothesize that this is caused by compounding errors in the inverse transition model. This hypothesis is supported by manually looking at trajectories generated by the inverse transition model. While they look reasonable for short horizons $T \leq 10$, compounding errors become significantly bigger for horizons $10 \leq T \leq 30$.

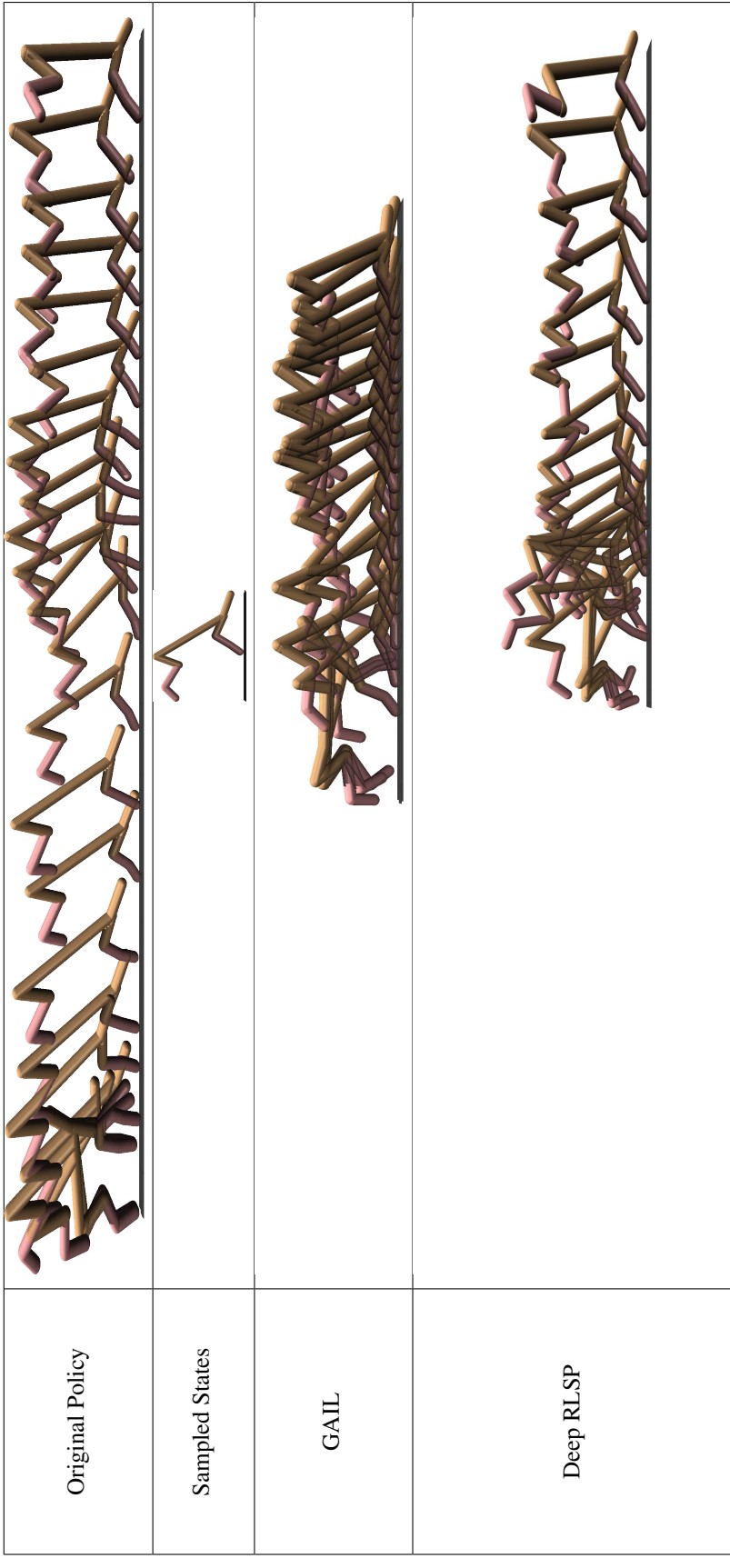

Figure 5: Deep RLSP learning the balancing skill from a single state. The first row shows the original policy from DADS, the second row shows the sampled state from this policy, the third row is the GAIL algorithm, and the last row shows the policy learned by Deep RLSP.

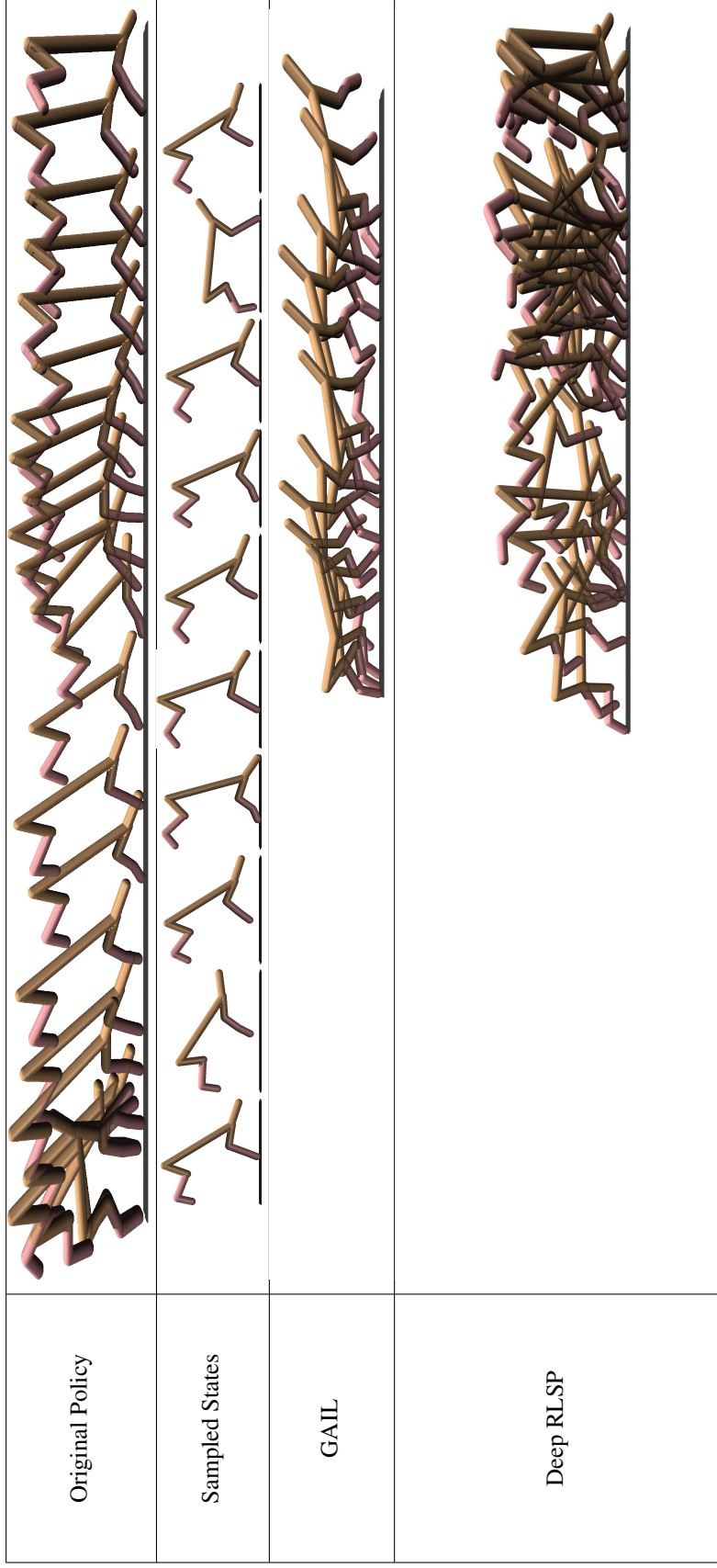

Figure 6: Deep RLSP learning the balancing skill from 10 states. The first row shows the original policy from DADS, the second row shows the sampled states from this policy, the third row is the GAIL algorithm, and the final row shows the policy learned by Deep RLSP.

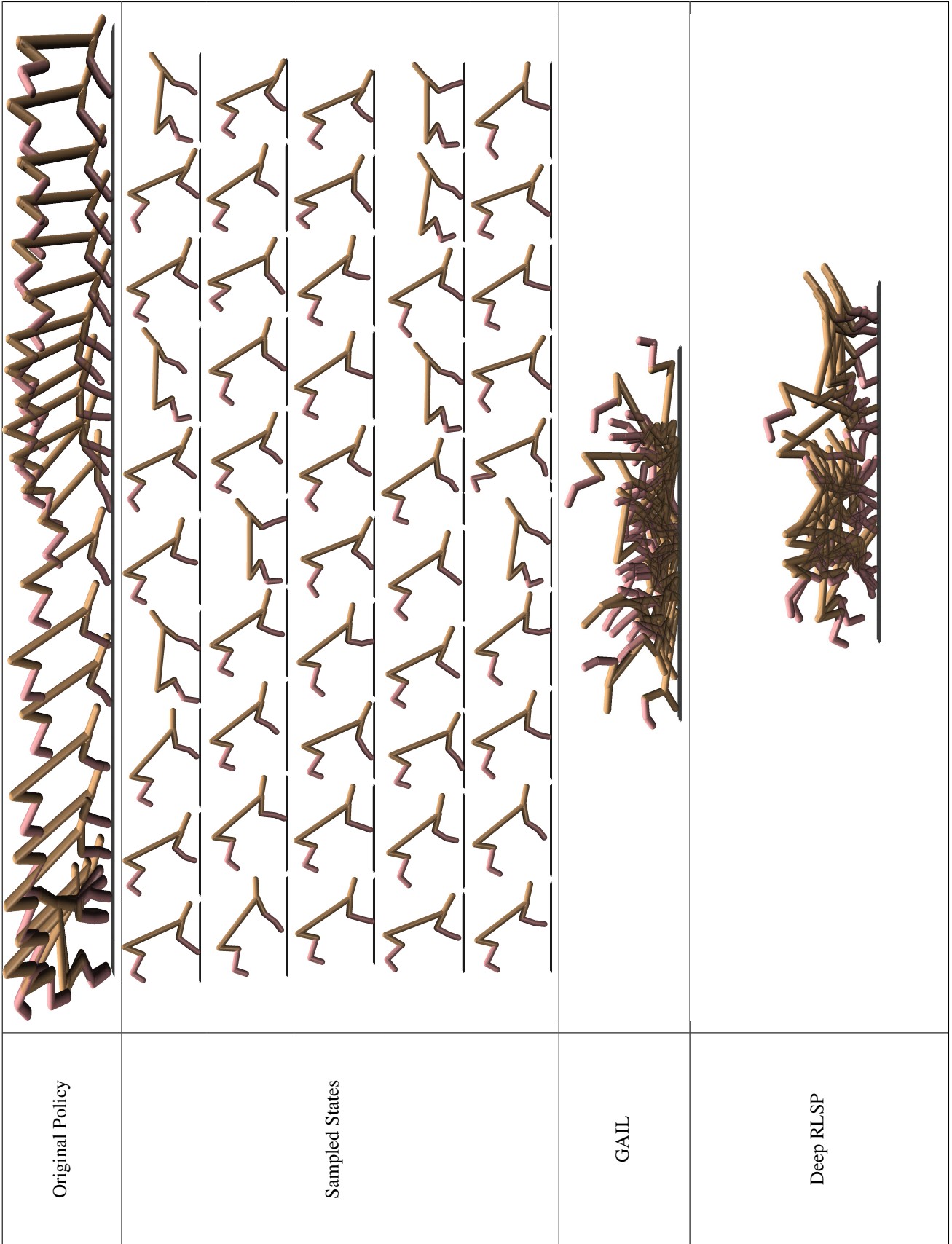

Figure 7: Deep RLSP learning the balancing skill from 50 states. The first row shows the original policy from DADS, the next five rows show the sampled states from this policy, the second to last row is the GAIL algorithm, and the last row shows the policy learned by Deep RLSP.

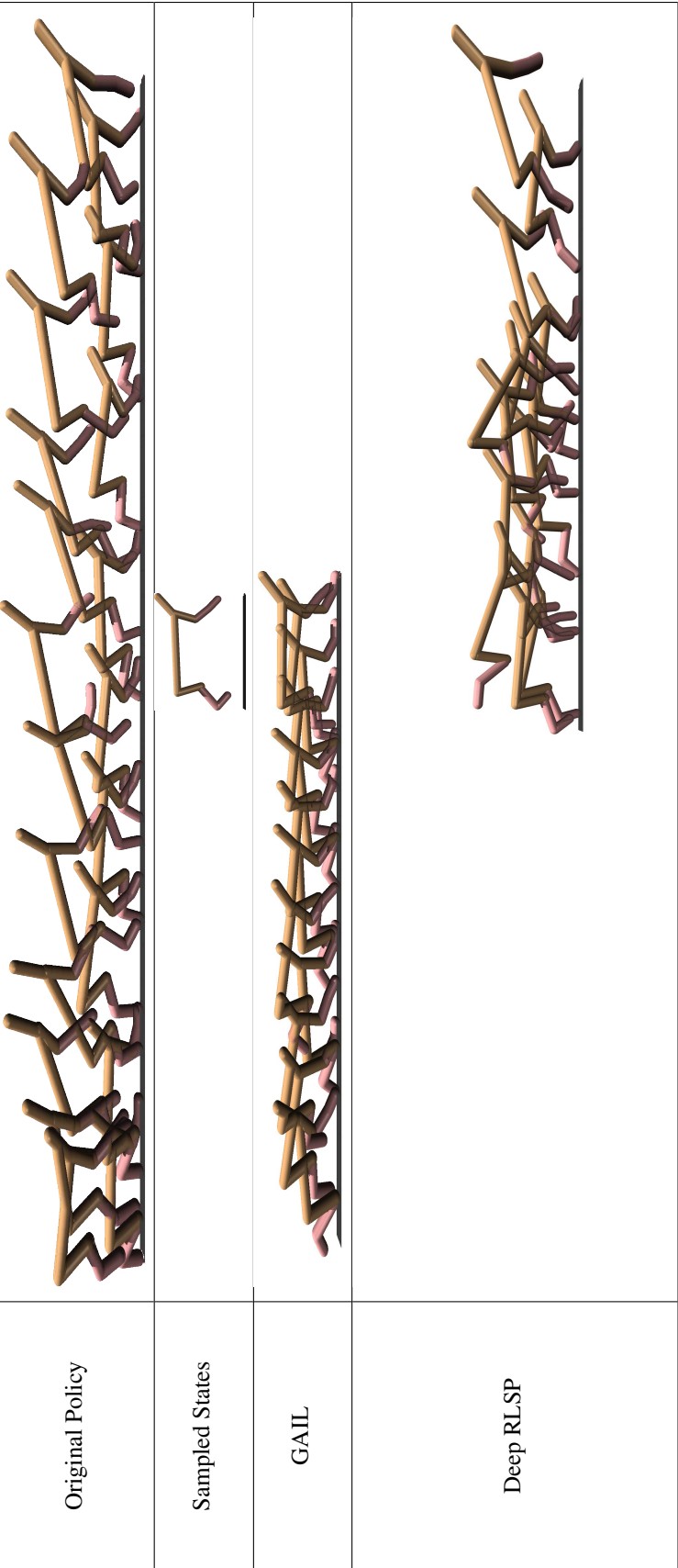

Figure 8: Deep RLSP learning the jumping skill from a single state. The first row shows the original policy from DADS, the second row shows the sampled state from this policy, the third row is the GAIL algorithm, and the last row shows the policy learned by Deep RLSP.

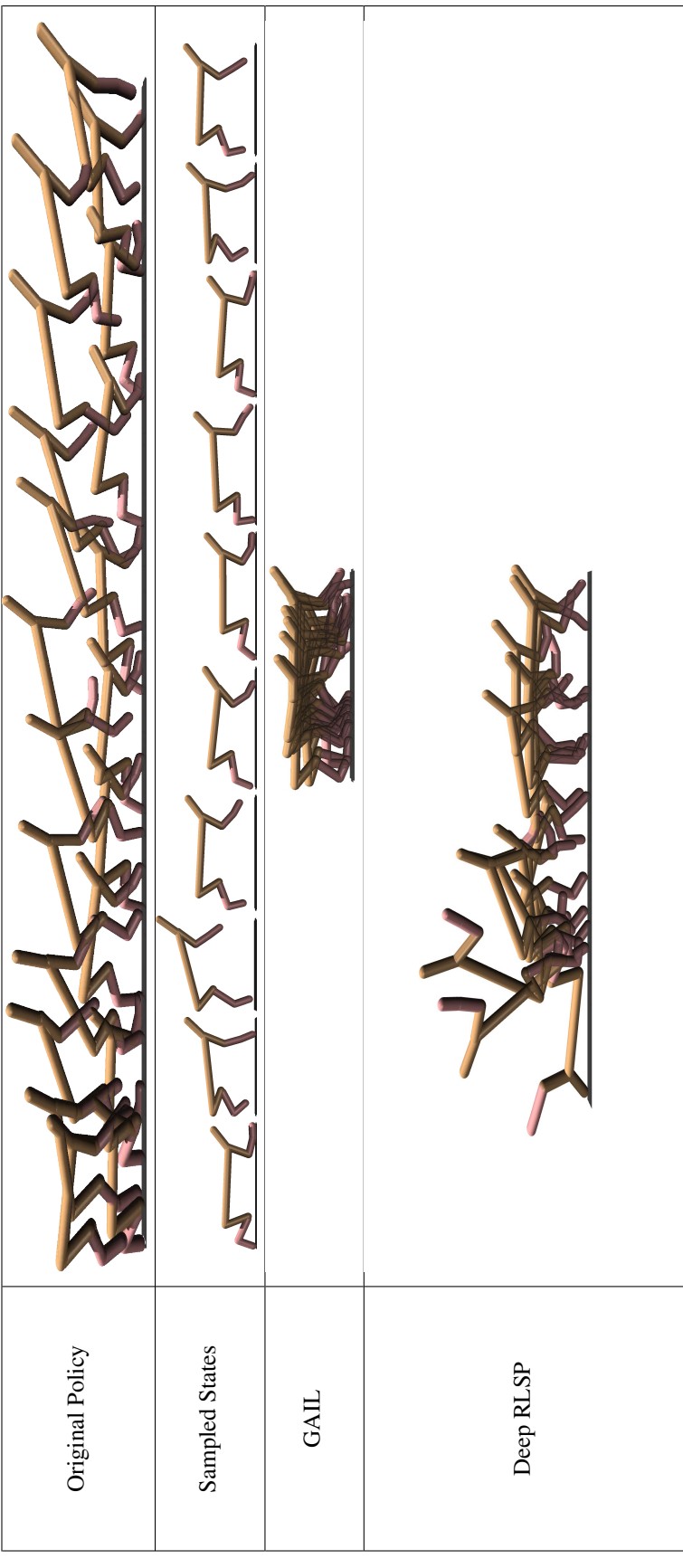

Figure 9: Deep RLSP learning the jumping skill from 10 states. The first row shows the original policy from DADS, the second row shows the sampled states from this policy, the third row is the GAIL algorithm, and the final row shows the policy learned by Deep RLSP.

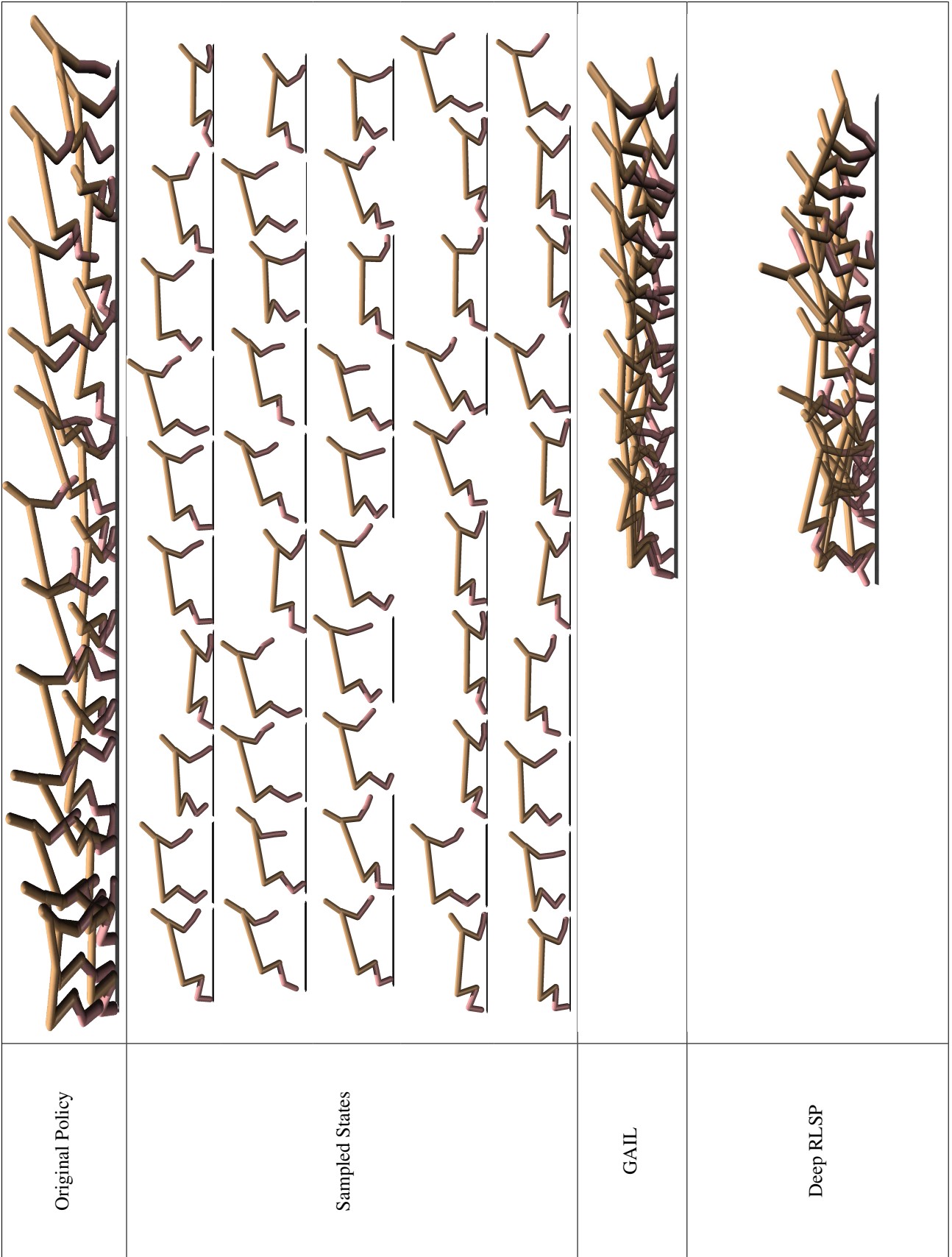

Figure 10: Deep RLSP learning the jumping skill from 50 states. The first row shows the original policy from DADS, the next five rows show the sampled states from this policy, the second to last row is the GAIL algorithm, and the last row shows the policy learned by Deep RLSP.

