# OpenReview forum: "Learning What To Do by Simulating the Past"
_ICLR.cc/2021/Conference — ICLR 2021 Poster_

### Official Review · AnonReviewer3 · 2020-10-28
**Cool idea but a bit limited in a number of ways. I did not find the empirical results fully convincing.**

**Rating:** 5
**Confidence:** 4

**Review:**

This paper introduces an algorithm, called deep reward learning by simulating the past (deep RLSP), that seeks to infer a reward function by looking at states in demonstration data. An example of this described in the paper is an environment with a vase: if demonstration data shows an intact vase in the presence of an embodied agent then breaking the vase is unlikely to be the intended behavior. Otherwise the vase would already be broken in the demo.

To achieve this, the paper assumes a Boltzmann distribution on the demonstration policy and a reward function that is linear in some pre-trained state features. The paper then derives a gradient of the log probability of a demonstration state. The gradient estimator involves simulating a possible past from a demonstration state (using a learned inverse policy and inverse transition function) and then simulating forward from the possible past (using the policy and a simulator) The gradient is then the difference between features counts from the backward and forward simulations.

The paper is generally clearly written and works on a crucial problem in reinforcement learning, namely how to specify a human preference without resorting to tedious reward engineering. Novel, scalable approaches to this problem would certainly be of interest to the ICLR community. The primary technical contribution of the paper is the derivation of the gradient estimator which is correct.

I find the idea of the paper very interesting and the results showing meaningful behavior emerge from a single demonstration are quite nice. However I think the paper is limited in a number of ways:
- It requires access to a pretrained state representation
- It requires access to a simulator of the environment which requires being able to reset the environment to arbitrary states. This seems quite limiting for real world applications. Worryingly, appendix D states that learning a dynamics model was attempted by the authors but failed to yield good results.
- I think the choice of evaluation environments is a little odd and simplistic. I think environments more aligned with the eventual application areas for a method such as Deep RLSP would make the paper much more compelling. Given the motivation of the paper, I think perhaps manipulation environments where a robot arm interacts with multiple objects could be an interesting choice.
- From the empirical results, it is not clear that Deep RLSP works substantially better than the simple average features baseline.

Overall I think the paper has the potential to be a good paper but could still be substantially improved and I'm leaning towards rejection.

Minor comments and questions for the authors:
- I'm curious how you choose the gradient magnitude threshold? Does Deep RLSP fail without the curriculum? Could you provide an ablation that shows the effect of using a curriculum?
- I would also be interested in an ablation of the cosine-similarity weighting heuristic.
- I think the phrase recent work in the abstract could use a reference.
- I'm a bit confused by the description of the environment suite by Shah et al. in appendix A, in particular the different rewards. Could you clarify and expand the description a bit?

---

> ### Author Response · Authors · 2020-11-22
> **Response**
>
> Thanks for the review! We are glad that you found the paper clear, the problem setting significant, the idea interesting, and the results nice. We hope to address your concerns below.
>
> > It requires access to a pretrained state representation. It requires access to a simulator of the environment which requires being able to reset the environment to arbitrary states.
>
> We argue that we have made a significant step forward from the current state of the art by reducing the amount of human supervision required. The current approach would be some combination of imitation learning (as with GAIL) or comparison learning (as in [1]). These approaches require a simulator as well as extensive human data. (One exception is behavioral cloning, which only requires human data, but requires a lot of it.) We remove the need for extensive human data, while still requiring the simulator; thus we argue that we have made substantial progress, while acknowledging that there is still the issue of how to get an appropriate simulator in the first place. Note that we learn the pretrained state representation from the simulator, so it is not an extra requirement beyond the simulator.
>
> We are also optimistic about the significance of our work because it appears as though learned dynamics models and feature functions are improving very quickly, arguably faster than reward learning. Dynamics and features can also be trained during development of an AI system, whereas reward should ideally be personalized to each user. (See also our response to R2.)
>
> We do require the ability to reset to a specific state. We agree that this is restrictive (though it has been used before, e.g. [2] and [3]). We wanted to avoid this by training a learned dynamics model, and then simulating forward trajectories using this model. While unfortunately this did not work well in our current experiments, we expect that dynamics models such as RSSMs will improve in the next few years such that they could be fruitfully applied here.
>
> > I think the choice of evaluation environments is a little odd and simplistic. [...] I think perhaps manipulation environments where a robot arm interacts with multiple objects could be an interesting choice.
>
> Yes, good environments were a major sticking point for us in doing this work. Typical environments are designed with RL in mind, and so do not usually make as much sense for Deep RLSP. A robot arm interacting with multiple objects could work well; we may try this in the future.
>
> > it is not clear that Deep RLSP works substantially better than the simple average features baseline.
>
> We’re not sure why you say this? It is true that for the tasks with rewards (Section 3.3), the two methods are comparable, but in the skill learning section AverageFeatures clearly fails badly -- it seems to always learn to run forwards.
>
> Incidentally, this suggests an explanation for the good performance of AverageFeatures in Section 3.3 -- perhaps rewards for “running” are particularly common for AverageFeatures, and so it does well on these sorts of tasks, but can’t learn other tasks like the skills of Section 3.4.
>
> In addition, AverageFeatures does not work on the gridworld environments. We have swapped Section 3.1 and 3.2 and added a note that AverageFeatures fails this sanity check.
>
> > I'm curious how you choose the gradient magnitude threshold? Does Deep RLSP fail without the curriculum?
>
> We tried out a few values and selected one that seemed to lead to stable training. We also advanced the curriculum after 10 iterations regardless of the gradient magnitude. So, we could get rid of the threshold and always run 10 iterations for each additional timestep, at the cost of at most 10x more compute.
>
> Deep RLSP does not work at all without the curriculum. We changed a sentence in Section 2.4 to make this clear. One way to think of it is that without the curriculum, we should expect Deep RLSP to have huge variance: it is much harder to make long backwards and forwards trajectories consistent with each other.
>
> > I would also be interested in an ablation of the cosine-similarity weighting heuristic.
>
> Great suggestion! In fact, it turns out the heuristic was not adding much at all. We have added a note to the main paper and added the details to Appendix C.
>
> > I'm a bit confused by the description of the environment suite by Shah et al [...]. Could you clarify and expand the description a bit?
>
> We’ve added details to Appendix A. Please let us know if we missed important details -- we weren’t sure what specifically you were interested in.
>
> References
>
> [1] Christiano, Paul F., et al. "Deep reinforcement learning from human preferences." Advances in Neural Information Processing Systems. 2017.
> [2] Ecoffet, Adrien, et al. "Go-explore: a new approach for hard-exploration problems." arXiv preprint arXiv:1901.10995 (2019).
> [3] Salimans, Tim, and Richard Chen. "Learning Montezuma's Revenge from a Single Demonstration." arXiv preprint arXiv:1812.03381 (2018).

---

> > ### Comment · AnonReviewer3 · 2020-11-23
> > **Thank you for your answers and clarifications!**
> >
> > I would like to thank the authors for their answers and clarifications in response to my review. I have read them and do not have any further questions at the moment. I will update my review after the end of the discussion period.

---

### Official Review · AnonReviewer1 · 2020-10-28

**Rating:** 7
**Confidence:** 2

**Review:**

The paper considers an approach for reward learning from a single frame which was developed for tabular environments and explicit dynamic programming, and extends it to more complex environments through deep RL.

Methodologically, the paper stays close to the ideas of RLSP, but instead of computing relevant quantities (optimal policy, forward and inverse dynamics) through explicit derivations, it suggests most exact steps can be replaced by leveraging deep learning, reinforcement learning and self-supervised learning. There are some minor degeneracy issues stemming from the extension (in particular in gridworld environment) which the authors can mostly solve.

While the paper does not have strong methodological novelty, it is well written, the approach is sensible and combines well with state of the art deep RL, and the results are certainly interesting. The authors do a good job providing ablations (some ablations working surprisingly well maybe suggests the methods is working for slightly different reasons than we may assume).



Questions:
- How is SAC(theta) computed? Is it a policy with its own parameters?
- The inverse dynamics model is a function of the current policy, which is changing over time. This usually causes 'tracking/lagging' issues, and here especially so because the inverse dynamics is presumably a very nonlinear function of the policy. Did you observe any such issues?





[1] Reward Learning by Simulating the Past (RLSP)

---

> ### Author Response · Authors · 2020-11-22
> **Overall in agreement**
>
> Thanks for the review! We’re glad you found that the method combined well the state of the art deep RL (this was indeed a key desideratum for us), and that the experiments and results were interesting. We respond to the questions below.
>
> > How is SAC(theta) computed? Is it a policy with its own parameters?
>
> Yes, that’s exactly right.
>
> > The inverse dynamics model is a function of the current policy, which is changing over time. This usually causes 'tracking/lagging' issues, and here especially so because the inverse dynamics is presumably a very nonlinear function of the policy. Did you observe any such issues?
>
> Yes, we did. The first thing we tried had the inverse dynamics “inside the loop”, that is, we continually retrained the inverse dynamics model as the policy changed, and indeed this led to a lot of stability issues. In the end we “solved” this by just not retraining the inverse dynamics -- in Algorithm 1 the inverse dynamics model is trained once at the beginning and then never trained while the policy is being updated. This significantly improves stability, and it turns out the fact that the model is being trained in a slightly incorrect way does not matter much in practice (at least for our experiments).
>
> (This was also mentioned in the last bullet point of Appendix D, now Appendix E.)
>
> > some ablations working surprisingly well maybe suggests the methods is working for slightly different reasons than we may assume
>
> Indeed, we also found this somewhat surprising. One thing to note is that Deep RLSP is the only algorithm that performs “reasonably” across all of the settings -- all the ablations and baselines fail quite badly on at least one setting. This suggests to us that the surprising success comes from some property of the environment, though we are not sure what. For example, perhaps “most” reward functions tend to incentivize the Cheetah to run, and so it is particularly easy for a method inferring a reward function to succeed well on the Cheetah task.

---

### Official Review · AnonReviewer2 · 2020-10-29

**Rating:** 5
**Confidence:** 2

**Review:**

This paper studies the question of learning rewards given only certain preferred or terminal states. I found the setting of the problem to be interesting but not clearly motivated or explained. For this reason, I am unable to recommend acceptance at this stage, due to major clarity issues. I am open to revisiting the recommendation based on author feedback.

(1) Can you explain the assumptions and setting a bit better? Do we only have access to terminal states from execution traces of an expert/human policy? I assume additional interactions with the environment are allowed?

(2) The derivation for equation (1) is unclear to me. I understand that $P(s_0 | \theta) = \sum_{\tau \in \Omega} P(\tau | \theta)$ where $\Omega$ are the subset of trajectories that terminate in state $s_0$. Since the probability is a sum of probabilities (instead of product), why does the gradient of log probability decompose into a linear combination of the trajectory gradients?

(3) The writing and notations are quite confusing and hard to follow. In particular, the negative time indexing makes a number of expressions counter intuitive. For example, let us take the expression in section 2.2:
- What are the contents in $\ldots$? Can you expand the equation further?
- Why is $P(a_t | s_{t+1}, a_{t+1}, \ldots s_0, \theta) = P(a_t | s_{t+1}, \theta)$? From a causality viewpoint, this seems counter intuitive since a future state should not influence current action. Is it actually a typo? If this instead has a filtering style interpretation, then shouldn't the decomposition be $P(a_t | s_t, s_{t+1}, \theta)$ and $P(s_t | s_{t+1}, \theta)$?
- It might be easier for parsing and understanding to have the product be $\prod_{t=-T}^{t=-1}$ instead of $\prod_{t=-1}^{t=-T}$.

(4) Based on an educated guess of the problem setting (per my understanding), an intuitive and perhaps simpler algorithm would be to learn a goal classifier using the provided {s_0} data, i.e. $P(s=\text{goal})$, as well as a forward dynamics model $P(s_{t+1}|s_t,a_t)$. Then, one can use the goal classifier as a reward for planning. Would this be an applicable algorithm for this setting? Is this related to the GAIL baseline?

Overall, I was unable to understand the problem formulation and setup, which makes it hard to appreciate the experiments. If the authors can better motivate the setting and improve writing clarity and notations during the rebuttal revision, I am happy to revise my score.

---

> ### Author Response · Authors · 2020-11-22
> **Text Clarifications**
>
> Thanks for the review! We’re sorry that the paper was hard to understand. We will try to address the concerns you raise here, but unfortunately due to space limitations we cannot put all of this discussion in the paper. We would especially welcome any additional feedback on which parts of this explanation were useful, so that we can try to incorporate them into the paper.
>
> > Can you explain the assumptions and setting a bit better?
>
> Here is a vision for one possible long-term goal: household robots. We can imagine that developers of household robots will equip them with a good understanding of the world, that allows them to e.g. wash dishes, fold laundry, tidy up a room, etc. Thus, developers could ensure that they have good dynamics models and feature functions, perhaps via domain randomization in simulation, or by collecting lots of data in the real world in a variety of environments, or something else.
>
> However, it is not clear how developers could give household robots an appropriate reward function -- everyone organizes their house differently; the robot should be able to be personalized to the specific user whose house they are working at. For example, perhaps some people prefer for shoes to be placed on a shoe rack, while others do not mind having the shoes in a jumbled mess near the front door, and others are happy to have people wear their shoes in the house. It would be nice if the robot could simply look at the current state of the house and determine which of these is the case. Our hope is that some algorithm like the one we present would be able to be applied in such a scenario. It is of course possible to have simpler solutions in cases like this, such as adding a simple subroutine to search for shoe racks and to use them if they are present; but the hoped-for benefit is that the robot could infer such preferences without the developers having to think about the existence of these preferences in advance.
>
> For the specific algorithm we present in this paper, we are assuming access to a simulator of the environment, that can be reset to specific states, in order to get information about the dynamics. We allow arbitrary amounts of interaction with this simulator.
>
> In terms of the human input, it becomes a little more complicated, since Deep RLSP is fundamentally modeling what the human does, but then using the result to influence the robot behavior. If the goal is to use the policy learned by Deep RLSP on the robot, then we are implicitly assuming that the human and robot have the same action space; in this case we would only have access to terminal states from an expert policy. However, if the goal is to use the _reward_ learned by Deep RLSP to train the robot, then we do not have to make this assumption: we can use Deep RLSP with a model of the _human’s_ action space to infer what the human cares about, and then use the reward to train the robot in a different environment simulator that models the environment with the robot’s action space. This is the sort of situation we hope to eventually handle, as illustrated in the household robot example.
>
> > an intuitive and perhaps simpler algorithm would be to learn a goal classifier using the provided {s_0} data [...] one can use the goal classifier as a reward for planning
>
> Yes, this would be a reasonable baseline in this setting. This is somewhat similar to the AverageFeatures algorithm, which effectively rewards the agent for being near the {s_0} data, though it is importantly different. It is more similar to the “deviation” baseline in the prior work by Shah et al, which can be thought of as a continuous nearest-neighbors based goal classifier.
>
> The main issue with such algorithms is that they can’t take into account information about the dynamics, which is often important. For example, in a room with a breakable vase, if you observe that the vase is unbroken, you can infer that the human has _never_ broken the vase (since breaking the vase is irreversible, which you know via dynamics information), and so you can strongly infer that it is bad to break the vase. However, if instead the vase could be easily repaired via magic superglue after being broken, then you should no longer infer so strongly that the vase must not be broken -- it’s possible that it’s fine to break the vase, and the human has broken the vase many times in the past but has just repaired it with superglue. The AverageFeatures baseline and the goal-based classifier would not be able to make these sorts of distinctions.
>
> This is better illustrated in Shah et al, but even in our experiments we see that the use of dynamics information is important in the skill learning experiments (Section 3.4).

---

> ### Author Response · Authors · 2020-11-22
> **Math clarifications**
>
> > The derivation for equation (1) is unclear to me.
>
> Here is the full derivation, copied from Appendix B of Shah et al (which that section is summarizing):
>
> $$\nabla_{\theta} \ln p(s_0 \mid \theta) = \frac{1}{p(s_0 \mid \theta)} \nabla_{\theta} p(s_0 \mid \theta)$$
> $$= \frac{1}{p(s_0 \mid \theta)} \sum\limits_{s_{-T:-1},a_{-T:0}} \nabla_{\theta} p(\tau_{-T:0} \mid \theta)$$
> $$= \frac{1}{p(s_0 \mid \theta)} \sum\limits_{s_{-T:-1},a_{-T:0}} p(\tau_{-T:0} \mid \theta)\nabla_{\theta} \ln p(\tau_{-T:0} \mid \theta)$$
> $$= \frac{1}{p(s_0 \mid \theta)} \sum\limits_{s_{-T:-1},a_{-T:-1}} \left( p(\tau_{-T:-1}, s_0 \mid \theta) \nabla_{\theta} \ln p(\tau_{-T:0} \mid \theta) \left( \sum\limits_{a_0} \pi_0(a_0 \mid s_0, \theta) \right) \right)$$
> $$= \sum\limits_{s_{-T:-1},a_{-T:-1}} p(\tau_{-T:-1} \mid s_0, \theta) \nabla_{\theta} \ln p(\tau_{-T:0} \mid \theta)$$
> $$= E_{\tau_{-T:-1} \sim p(\tau_{-T:-1} \mid s_0, \theta)} \bigg[ \nabla_{\theta} \ln p(\tau_{-T:0} \mid \theta) \bigg].$$
>
> We have added a citation to Appendix B of that paper.
>
> > It might be easier for parsing and understanding to have the product be $\prod_{t=-T}^{t=-1}$ instead of $\prod_{t=-1}^{t=-T}$
>
> Good point, we have made this change. (The original way is more in the spirit of “simulating the past”, as you start with the present and then move backwards in time, but in hindsight we agree that it is more confusing than edifying.)
>
> > What are the contents in …?
>
> Let us assume that the human has acted for three timesteps, and we’ve switched to the -T to -1 ordering suggested above. Then the equation would be:
>
> $p(\tau_{-3:-1} \mid s_0, \theta) = p(s_{-3} \mid a_{-3}, s_{-2}, a_{-2}, s_{-1}, a_{-1}, s_0, \theta) p(a_{-3} \mid s_{-2}, a_{-2}, s_{-1}, a_{-1}, s_0, \theta) p(s_{-2} \mid a_{-2}, s_{-1}, a_{-1}, s_0, \theta) p(a_{-2} \mid s_{-1}, a_{-1}, s_0, \theta)  p(s_{-1} \mid a_{-1}, s_0, \theta) p(a_{-1} \mid s_0, \theta)$
>
> > Why is $P(a_t \mid s_{t+1}, a_{t+1}, \dots s_0, \theta)=P(a_t \mid s_{t+1}, \theta)$?
>
> This is neither a causality nor a filtering interpretation. We don’t know of any specific interpretation that makes this intuitive, but it is a fact that can be deduced from the Bayes Net describing a Markov Decision Process. This Bayes Net is shown below in the case where there are only 2 actions (where we have elided θ, which is a parent to every other node in the graph):
>
>     s0 --> s1 --> s2
>     |     ^ |    ^
>     v    /  v   /
>     a0---  a1---
>
> We can show that $P(a_0 \mid s_1, a_1, s_2, \theta) = P(a_0 \mid s_1, \theta)$ in this Bayes Net (one way to verify is to use the d-separation algorithm). The key idea is that for any action $a_t$, the next state $s_{t+1}$ along with $\theta$ together screen off all information about the future (they form a cut in the graph).

---

### Official Review · AnonReviewer4 · 2020-10-29
**An original setting, and an interesting and promising approach.**

**Rating:** 7
**Confidence:** 4

**Review:**

This paper introduces an imitation learning algorithm that can take a single state from an expert's trajectory in order to infer the goal of the task. The idea is to train a reward function that explains both the past trajectory and the futur trajectory from that state, assuming that the very goal was that state (hence, the assumption of larger rewards in the past than in the future induced by the gradient).

The idea makes sens and is well explained. The results are encouraging and support the method.

I also believe this work could be impactful in the explanation domain, as it provides an answer to the question "why this state?". Therefore, outside simple imitation, it could even apply to self-imitation or credit-assigment in direct RL settings. The only weakness would be the required heaviness of the implementation: one need to implement and train all the models (direct and inverse dynamics and policies, auto-encoders for features etc).

However, I would ask for some clarifications:
- What idea that could help the reader to understanding equation (2) (without going through the appendix of the cited paper)?
- In equation (3), is \tau' independent of \tau_-T:-1, so the right-hand term (in red) can be moved outside the blue expectation? (so the simple difference in the last line of the algorithm would nicely appear?)
- What is encoded and decoded by VAE? (I guess the states, but this is not explicitly told)

Also, I would have expected a discussion regarding the environmental limitation of this approach, for example, when a state can be completely misleading, or at least not providing any information (for example, the initial state of an environment).

---

> ### Author Response · Authors · 2020-11-22
> **Response**
>
> Thanks for the review! We’re glad you found the idea interesting and our results encouraging. We respond to the specific individual points below.
>
> > The only weakness would be the required heaviness of the implementation
>
> We agree, the implementation is currently rather complex. Our hope is that this can be made simpler in the future by using a recurrent state space model (RSSM), as in [1]. A bidirectional RSSM would package the feature function, dynamics model, and inverse dynamics model into a single model. Unfortunately, we had trouble getting even a unidirectional RSSM to work stably -- we think this is just because RSSMs are in their infancy, and that in a few years when they work better we will be able to use them to simplify implementation.
>
> Clarifications:
>
> > What idea that could help the reader to understanding equation (2) (without going through the appendix of the cited paper)?
>
> Intuitively, the first (green) term increases the reward of the observed trajectory, the second (red) term decreases the reward of the trajectories that we would take under the current reward function, and the last (purple) term is a correction term that accounts for the fact that the agent does not control the dynamics, and so for every $(s_t, a_t, s_{t+1})$ triple, the agent “could have” seen a different triple $(s_t, a_t, s’_{t+1})$ if we had sampled differently from the transition dynamics. (If the environment is deterministic, then the last term is always zero.)
>
> We discuss this to some extent after Equation (3). We have moved some of the discussion earlier to show that it also applies to Equation (2).
>
> > In equation (3), is \tau' independent of \tau_-T:-1, so the right-hand term (in red) can be moved outside the blue expectation? (so the simple difference in the last line of the algorithm would nicely appear?)
>
> This is almost correct. The one dependency is that $\tau’$ starts from state $s_{-T}$, which is part of $\tau_{-T:-1}$. So we could instead say that $\tau’$ is independent of $\tau_{-T:-1}$ given $s_{-T}$.
>
> This means that the red term cannot actually be moved outside the blue expectation -- we do have to first do the backwards simulation (in blue) in order to get s_{-T}. However, then once we have s_{-T}, we can simulate $\tau’$ without worrying about the blue expectation. The algorithm exactly mimics this structure.
>
> > What is encoded and decoded by VAE? (I guess the states, but this is not explicitly told)
>
> Yes, it is the states. We have added a sentence to clarify.
>
> > I also believe this work could be impactful in the explanation domain
>
> We hadn’t thought about the application of the method to explanations -- thinking of the method as a way of asking the question “why this state” seems particularly interesting. Did you have a concrete scenario in mind where it seems like you could benefit from applying Deep RLSP?
>
>
> References
> [1] Hafner, Danijar, et al. "Dream to Control: Learning Behaviors by Latent Imagination." International Conference on Learning Representations. 2019.

---

### Decision · Program_Chairs · 2021-01-07
**Final Decision**

**Decision:**

Accept (Poster)

**Comment:**

First as a procedural point, the paper got 7, 7, 5, 5. AnonReviewer3 gave it a 5, but seemed satisfied by the discussion and promised to raise their score. They did not do so, but I must interpret their last messages as indicating they now support the paper. AnonReviewer2, the other 5, had some concerns that other reviewers seem to have helped address during rebuttal. They did not update their score, but were happy to leave their certainty low and defer to other reviewers' recommendation. As such, although the average score looks low in the system, the paper is of an acceptable standard according to reviews.

The paper adapts a method from tabular RL to Deep RL, allowing (as the title aptly says), agents to learn What to do by simulating the past. Reviewers speaking in support of the paper found that the paper was clear and sound in its evaluation, providing interesting results and a useful and reusable method. It is my feeling that after discussion, the case for the paper has been clearly made, and in the absence of any strong objections from the reviewers, I am happy to go with the consensus and recommend acceptance.